# The Effects of Jet-Milling and Pulsed Electric Fields on the Preservation of Spinach Juice Lutein Contents during Storage

**DOI:** 10.3390/foods13060834

**Published:** 2024-03-08

**Authors:** Si-Yeon Kim, Yeong-Geol Lee, Hye-In Ju, Ji-Hee Jeon, Se-Ho Jeong, Dong-Un Lee

**Affiliations:** 1Department of Food Science and Technology, Chung-Ang University, Anseong 17546, Republic of Korea; asdfisiof1@cau.ac.kr (S.-Y.K.); dldkqm@cau.ac.kr (H.-I.J.); wjswlgml901@cau.ac.kr (J.-H.J.); mrr003@cau.ac.kr (S.-H.J.); 2Devotion Foods Company, 15, Pildong-ro, Jung-gu, Seoul 04624, Republic of Korea; aagxo3159@cau.ac.kr

**Keywords:** jet-milling, spinach, lutein, pasteurization, pulsed electric fields

## Abstract

This study aimed to investigate the effects of jet-milling on the lutein extraction contents of spinach powder (SP), as well as the effects of pulsed electric field (PEF), as a non-thermal pasteurization technology, on the preservation of spinach juice (SJ) lutein contents. SP particles were divided into SP-coarse (Dv50 = 315.2 μm), SP-fine (Dv50 = 125.20 μm), and SP-superfine (Dv50 = 5.59 μm) fractions, and SP-superfine was added to SJ due to its having the highest contents of lutein extract. PEFs and thermal treatment were applied to evaluate the effects of preserving the lutein content of PEF during storage (25 days). The juice was then designated as untreated (no pasteurization), PEF-1,2 (SJ treated with PEF 20 kV/cm 110 kJ/L, 150 kJ/L), or Thermal-1,2 (SJ treated with 90 °C, 10 min and 121 °C, 15 min). The sizes and surface shapes of the superfine SP particles were more homogeneous and smoother than those of the other samples. SJ made with SP-superfine and treated with PEF had the highest lutein content and antioxidant activities among the group during storage. A complex of jet-milling and PEF could have great potential as a method to improve the lutein contents of lutein-enriched juice in the food industry.

## 1. Introduction

Spinach (*Spinacia oleracea*) is commonly consumed as a lutein-rich leafy vegetable in western countries. It also has glycosylates, chlorophylls, vitamins, minerals, and bioactive compounds [1]. Spinach, a food source of lutein, is often consumed as a pharmaceutical, health supplement, extract, and functional beverage [2]. Lutein is the representative macular carotenoid pigment that prevents numerous eye diseases due to its different chemical structure compared to carotenes [3]. Generally, carotenoids are composed of 40-carbon skeletons with two 20-carbon precursors conjugated with double bonds, and carotenes are subgroups of hydrocarbons that consist of carotenoids [4]. Lutein is a subgroup that has two hydroxyl groups bound with ionone rings of the terminal on both sides. This chemical structure also gives it polar and hydrophilic properties and improves its reaction with oxygen [4]. However, the chemical changes, instability, and low bioaccessibility and bioavailability of lutein are the main limitations of its application in the food industry because its processing conditions include high temperatures, the presence of oxygen and light, and extreme pH values (below 4, above 8.0), which destroy the integrity of lutein [5].

For this reason, preserving lutein contents during food processing and increasing its bioaccessibility are required [6]. Furthermore, to preserve lutein contents during food processing, extraction is mainly used, but the thermal process should also increase the food’s shelf life and decrease the risk of microbial contamination. However, the lutein content in the extract dramatically decreases due to the thermal process of pasteurization. Therefore, to solve this problem and improve the extraction yield of lutein, using an increased quantity of lutein extraction and non-thermal pasteurization technologies is essential.

A previous study showed that the particle size reduction in raw materials preserves and improves the yield of lutein extraction [7]. Jet-milling is a prospective superfine grinding technology that produces micro- and nanoparticles (<40 μm) using highly pressurized air streams while preserving hydrophilic and hydrophobic molecular structure characteristics [8,9]. Another previous study showed that the particle size reduction in raw materials preserves and improves the yield of lutein extraction; the lutein extracted from superfine marigold powder (Dv50 = 10 μm) was approximately 40% more than that extracted from coarse marigold powder (Dv50 = 315 μm). Therefore, jet-milling technology has been widely studied and applied in pharmaceutical manufacturing [10], barley and rye flours [11], sargassum fusiform [12], and wheat flour [13] to improve the extraction yield and its physicochemical, mechanical, and rheological properties. However, there is no further research on the extraction of lutein from spinach powder using jet-milling except for our previous research [7].

Pulsed electric fields are a type of non-thermal pasteurization technology that uses intense electric fields with a short duration (milliseconds to microseconds), which preserves the functional compounds more than thermal technology does [14]. PEF treatment initiates the repulsion between change-carrying molecules and forms pores on the cell membrane, which affects the microbial inactivation, depending on the raw materials’ cellular structure, length, duration of field, and number of pulses [15]. However, excessive treatment with PEFs increases the temperature during pasteurization; therefore, optimizing the condition of the PEFs is crucial [16]. Previous studies investigated the effects of PEFs on pasteurized orange juice [17], apple juice [15], and fruit juice [18], with no further degradation of the bioactive compounds. However, there has been no research performed including the complex technology of jet-milling with PEF treatment on spinach juice to preserve the lutein content.

The aim of the present study is to investigate the application of jet-milling and PEFs to produce spinach juice and increase and preserve its lutein content. To compare the effects of particle size reduction, coarse, fine, and superfine spinach powder was prepared, and we investigated the resultant particle size distribution, hydration and oil binding capacity, lutein contents, and microbial activities. After that, PEF-1 (20 kV/cm for 110 kJ/L) and PEF-2 (20 kV/cm 150 kJ/L) were treated to compare their pasteurizing effects on the preservation of lutein content compared to thermal technologies (Thermal-1—95 °C for 10 min, Thermal-2—121 °C for 15 min).

## 2. Materials and Methods

Fresh whole spinach leaves were purchased from a nonghyup in Anseong, Republic of Korea. Then, spinach without stems was dried for 48 s at 50 °C in a forced-convection drying oven (SFC-203, Shinsaeng, Paju, Republic of Korea) and ground using a grinder. Approximately 8–10 g of spinach was selected, and 20–30 leaves per group were used for the experiment. Raw spinach powder (SP) was passed through two sieve levels (150 µm and 63 µm testing sieve, Nonaka Rikaki, Chiyoda, Tokyo, Japan) using a sieve shaker (EML200, Haver and Boecker, Oelde, Germany). The SP was prepared with sieving using a 150 µm sieve, and the powder which could not pass through the 150 µm was collected and labeled SP-coarse. SP-fine was collected using a 63 µm sieve, and the powder which could not pass through the 63 µm was collected and labeled SP-fine. SP-superfine was obtained through the fluidized-bed jet mill (CGS 10, NETZSCH, Selb, Germany), and the conditions for the jet-milling were 7 bar of milling pressure and 12,000 rpm for the classifier. All SP was stored at −18 °C until use in the experiment.

### 2.1. Determination of Particle Size Distribution

The particle size of the SP used in the experiment was determined in triplicate by a particle size analyzer (Mastersizer 3000, Malvern Instrument Limited, Malvern, UK). The size distribution of each sample was applied using the dry method and a diffraction laser, and the operating conditions were set with particle absorption index (0.1), dispersant refractive index (1.00), laser obscurity 1.45%, and particle refractive index (1.53).

### 2.2. Physical Properties of Spinach Powder (SP)

To investigate the physical properties of SP, the bulk density, tap density, and carr index were measured, and the procedure followed the method in [19] with some modifications. SP samples were poured into the cylinder and the volume was filled to 50 mL; then, the cylinder weights with 50 mL of powder were measured. The bulk density (g/mL) was calculated as the injection weight of the sample (g) divided by the sample volume (mL). For tap density measurements, SP samples were poured into the cylinder and the volume was filled to 50 mL. Then, the cylinder was tapped strongly 100 times to fill any of the remaining space for homogeneity. The cylinder weights and sample volume of powder were measured. The tap density (g/mL) was calculated as the injection weight of the sample (g) divided by the sample volume (mL). The flowability/compressibility of spinach powder samples was calculated according to Shah, Tawakkul [20]. The water holding capacity (WHC) and Swelling capacity (SC) were measured and calculated as described in Zhang, Song [21], followed by Equations (1) and (2). The oil holding capacity (OHC) was determined following the method of Choi and Ma [22], with some modifications, and calculated via Equation (3).
WHC (g/g) = (Weight of sample (g))/(Weight of sediment (g))(1)
SC (mL/g) = (Volume occupied by sample (mL))/(Weight of dry sample (g))(2)
OHC (g/g) = (Weight of sediment (g) − Dry weight (g))/(Weight of sample (g))(3)

### 2.3. Scanning Electron Microscope (SEM)

The microstructures of SP were examined via scanning electron microscope (S-3400N, Hitachi High-Technologies Co., Tokyo, Japan). The sample was freeze-dried, fixed with carbon tape, and plated into aluminum specimen stubs. The pretreated samples were coated with platinum–lead (Pt-Pb) and then observed in a vacuum state. Then, the microstructure of SP was observed at 40×, 100×, and 200× magnification.

### 2.4. Spinach Juice (SJ) Preparation and Pasteurization

To prepare the spinach juice (SJ), SP-superfine was extracted using ethanol (SP-superfine:ethanol = 1:60) and centrifuged at 10,000 rpm for 10 min; then, the supernatant was collected. The supernatants from extracts were evaporated under vacuum at 40 °C; then, the dried extract was dispersed in distilled water to attain a value of 7 brix (%) and was used to make juice. The model juice with spinach extract was prepared with added sugar (20%), citric acid (0.03%), guar gum (0.01%), and spinach extract, and each sample was added with 4 mL of spinach extract per 200 mL of distilled water. For each group, 5 samples were measured after being placed in a glass bottle and homogenized for 3 min at 10,000 rpm using a homogenizer (T18 ultra-turra, IKA, Staufen, Germany) before the experiment.

The thermal treatment conditions of SJ were as follows; the thermal treatment was performed in two conditions (Thermal-1—95 °C for 10 min using a water bath (WB-11, Daihan, Osan, Republic of Korea); Thermal-2—121 °C for 15 min using an autoclave (BF-AC45, BNF, Gimpo, Republic of Korea)); after the treatment process, the SJ was instantly cooled in an ice bath until 4 °C was reached. For PEF treatment (pulsed electric fields, 5 kW, DIL, Quakenbruck, Germany), a 5-kW pulse generator (HVP-5; DIL, Quakenbrück, Germany), a continuous chamber with a 10 mm diameter tubular structure, and a pulse of the bipolar square type were used. The input and output temperature were 35 °C, and the flow rate of the SJ was 35 L/h, which is controlled using a peristaltic pump. The SJ was treated with 20 kV/cm 110 kJ/L and 20 kV/cm 150 kJ/L. After PEF treatment, SJ was cooled in an ice bath until 4 °C was reached and stored at different storage temperatures of 15 °C, 25 °C, and 35 °C for 25 days. The measurements were performed at 5-day intervals.

### 2.5. Determination of Microbial Activity

The reduction of microbial activity of SJ was evaluated by the number of total aerobic bacteria (TAB), which was measured using 3M petrifilm aerobic counts (PAC, 3M, Hudson, NY, USA). All samples were conducted via 10-fold serial dilution and incubated at 35 °C ± 1 for 48 h. After incubation, the number of total aerobic bacteria was counted.

### 2.6. Analysis of Lutein Contents

To extract the lutein in SP, 1 g of the sample was mixed with 50 mL of ethanol using a shaking incubator at 40 °C and 300 rpm for 30 min. The extract was centrifuged at 5000 rpm for 10 min; then, the supernatant was collected as described in Derrien, Badr [23]. To prepare the extract of SJ, 5 mL of the sample was mixed with 30 mL of hexane under a shaking incubator at 40 °C and 300 rpm for 3 h then centrifuged at 5000 rpm for 10 min. The supernatant was collected, and the residue was centrifuged twice more to obtain a decolorized residue. The supernatants from SP and SJ were evaporated under vacuum at 40 °C. The dry extract was dissolved in mobile phase A and filtered through a 0.2 μm syringe filter before HPLC analysis. The high-performance liquid chromatography (HPLC) system (LC-4000 series, Jasco, Tokyo, Japan) was equipped with a UV detector set at 450 nm, a C30 YMC column (250 × 4.6 mm id., 5 mm), a pump, and an autosampler maintained at a constant temperature of 35 °C, controlled by ChromNAV software. V.2.0. Isocratic conditions were held for 40 min, followed by a 2% methyl–tertbutyl ether (MTBE) to 60% of B, held for 25 min, then by a subsequent 5 min linear gradient to return to 2% MTBE, isocratic for 5 min, and the total run time was 35 min.

### 2.7. Determination of Antioxidant Activities

To evaluate the antioxidant activity of SP and SJ, ABTS and DPPH assays were used. For both ABTS and DPPH assays, the extract of SP and SJ samples were diluted appropriately with ethanol or water, and the ascorbic acid was used as a positive control.

For the ABTS assay, the procedure followed the method of Arnao, Cano [24] with some modifications. The extract of SP and SJ was diluted to 10 mg/mL and used as a sample solution. The 7.4 mM ABTS solution and 2.6 mM potassium persulfate solution were prepared in equal quantities to react for 24 h at room temperature in the dark. Then, they were diluted by water to obtain an absorbance of 0.7 units using the spectrophotometer at 734 nm. The 180 μL of solution was added to 20 μL of sample solution and allowed to react for 30 min at room temperature in the dark. Then, the absorbance was measured at 734 nm using a spectrometer (Spectramax 190 Microplate Reader, San Jose, CA, USA).

For the DPPH assay, the procedure followed the method of Blois [25] with some modifications. A stock solution of 0.2 mM DPPH in 99% ethanol was prepared. The solution was diluted by ethanol to obtain an absorbance of 1.0 units at 517 nm. The 100 μL sample solution was added into the 100 μL solution and allowed to react for 30 min at room temperature in the dark. Then, the absorbance was measured at 517 nm using a spectrometer (Spectramax 190 Microplate Reader).

## 3. Results and Discussion

### 3.1. Particle Properties of Differently Sized SP

The particle size distribution of SP is presented in Table 1 and Figure 1. The Dv50 results of SP-coarse, SP-fine, and SP-superfine were 315.20 ± 3.37 µm, 125.20 ± 1.30 µm, and 5.59 ± 0.24 µm, respectively. D[4,3] also showed similar values of 329.00 ± 6.44 µm, 143.6 ± 2.07 µm, and 6.93 ± 0.24 µm, respectively. The specific surface areas of SP-coarse, SP-fine, and SP-superfine were 97.19 ± 4.11 m^2^/kg, 334.26 ± 11.33 m^2^/kg, and 8671.20 ± 414.26 m^2^/kg, which correlate with the results of particle size reduction. Dv10, Dv50, and Dv90 indicate the equivalent diameters at cumulative volumes of 10%, 50%, and 90%, respectively, and as the decrease in the value of D[4,3] indicates that the mean of the volume-weighted diameter decreased, an increase in the value of the span indicates that the particle has a narrow and homogenous particle size distribution [19]. Zhang and Song [21] reported that the jet-milled mushroom powder has a higher surface area than the higher particle size group. Also, similar results were found in [19]; the value of Dv50 and D[4,3], which were treated via jet-mill, dramatically decreased compared to coarse and fine soybean flour. Regarding the results of particle-size distribution, jet-milling effectively reduces the size of the particle and increases the homogenous particle size distribution, which influences the physical properties of the powder [26].

The bulk density of SP-coarse, SP-fine, and SP-superfine were 0.44 ± 0.00 g/mL, 0.44 ± 0.00 g/mL, and 0.22 ± 0.00 g/mL, which increased as the particle size increased. A significant decrease in the bulk density and tap density was observed in SP-superfine compared to SP-coarse and SP-fine. The Carr Index of SP-superfine was 43.47 ± 0.49%, which means it is a fluid powder with poor fluidity. The Carr Index of SP-fine and SP-coarse were 24.27 ± 0.59% and 13.65 ± 0.28%, respectively, showing better fluidity than SP-superfine. A Carr Index value of 0~20% indicates good fluidity of the powder due to higher moisture content and porosity [27]. Generally, lower density and cohesive energy correlate with an increase in viscosity than higher density due to the stronger interaction forces [28]. However, in this study, SP-superfine showed poor flowability, and similar research observed that the flowability gradually decreased depending on the particle size reduction due to the agglomeration, and these properties correlated with an increase in surface area [29]. Moreover, Xu, Liu [30] reported that the decrease in the contents of cellulose induced by the breakdown of the cellular structure during jet-milling could influence the viscosity and flowability of the superfine powder. In this research, jet-milling could significantly reduce the particle size and effectively crush the SP to a superfine scale of less than 10 µm, but this suggests that the fluidity of the juice could have a lower viscosity compared to the coarse and fine particles due to the agglomeration and the extremely higher specific surface area.

### 3.2. Hydration Properties and Oil Holding Capacity (OHC) of Differently Sized SP

The WHC (water holding capacity) and SC (swelling capacity) of SP-superfine showed the lowest value among the group. Also, the OHC of SP-coarse, SP-fine, and SP-superfine were 2.86 ± 0.01 g/g, 2.51 ± 0.03 g/g, and 2.30 ± 0.04 g/g, which decreased as the particle size increased. In agreement with Gong and Huang [31], as the particle size decreased, the WHC and OHC decreased due to the hydrogen bond cleaved by the crushing process; thus, the contents of insoluble cellulose decreased more than the coarse and fine mushroom powder. Ming, Chen [32] reported that the long cellulose chain may break into the short chain, and disruption of the proteins and polysaccharide structure leads to a decrease in WHC and OHC. However, rice and barley, which were milled in a jet-mill, showed higher WHC due to the damaged starch, in which more water was observed than in intact starch [33]. Also, Zhong, Zu [34] reported that superfine grape peel powder showed higher solubility due to the larger surface area, thus accelerating the dissolution time. Therefore, the hydration properties and OHC could be changed by the chemical properties of ingredients. In this research, the reason that the SP-superfine showed lower WHC and OHC may be due to the destroyed cellulose structure.

### 3.3. Lutein Contents and Antioxidant Activities of SP

Table 2 shows the lutein contents and antioxidant ability of the SP sample. The lutein contents of SP-coarse, SP-fine, and SP-superfine were 1.59 ± 0.02 mg/g, 1.70 ± 0.10 mg/g, and 2.09 ± 0.13 mg/g of powder, respectively. The lutein content of SP-superfine was about 30% higher than that of SP-coarse. These observations indicated that as the particle size decreases, the lutein content increases. The ABTS assay results for SP-coarse, SP-fine, and SP-superfine were 56.56 ± 2.01%, 55.31 ± 0.31%, and 65.01 ± 1.25%, respectively. The DPPH radical scavenging activity of SP-coarse, SP-fine, and SP-superfine were 39.36 ± 1.94%, 32.67 ± 1.36%, and 36.82 ± 1.65%, respectively. The ABTS assay revealed higher antioxidant ability as the particle size decreased. Similar results were found by Zhong, Zu [34]; in their study, superfine pomegranate peel powder showed the highest release of polyphenol and flavonoid contents compared to coarse and fine powder at the same time. Moreover, as the particle size of the barley decreased, the total phenolics content also decreased, as did the antioxidant capacity due to the higher aggregation of superfine deffated soybean flour particles [11]. Also, as the particle size reduced, the polysaccharides with antioxidant activity extracted more than the larger particles [35]. However, an excessive milling process dramatically increases the surface area, which can react with oxygen, and the antioxidant compounds’ oxidation of raw materials increases; thus, functional compounds decrease, which acts as an antioxidant [36]. In this study, the jet mill effectively breaks the cell wall, which could accelerate the release of the intracellular compounds due to the high pressure and shear force during the milling process, and the higher porosity of the particles lead to the acceleration of the extraction of lutein.

### 3.4. Microstructure of Differently Sized SP

The micrographics are shown in Figure 2. At the same magnification, the particle size of SP-superfine was significantly smaller than the other SP samples. The surface structure of SP-coarse and SP-fine showed a larger and relatively more irregular and intact structure than that of SP-superfine. The changes in the microstructure induced by the mechanical force of the jet mill, causing the significant increase in the surface area, gives agglomeration properties of the superfine powder. Similar results were found in Gong, Huang [31]; they found that superfine mushroom particles showed smaller, more homogenous, and more agglomerated powder than coarse (>60-mesh) and 100-mesh particles. Also, Zhao, Zhu [37] showed that superfine red grapes powder showed an amorphous domain caused by the breakdown of intermolecular bonds, and its morphology properties correlate with the solubility and enzymatic hydrolysis. Muttakin, Kim [19] reported that the jet mill induces the collision between the particles due to the high-pressure airflow; thus, smaller, non-porous, and more-solid particles are produced.

### 3.5. Optimization of Lutein Extraction Conditions

The results of the lutein content extracted by using different concentrations of ethanol are presented in Figure 3. With the increase in the ethanol concentration from 40% to 80% in the extraction system, the lutein content increased from 0.24 mg/g to 2.13 mg/g of powder. The content of lutein increased proportionally with the ethanol/water ratio, in line with the lipophilic characteristic of lutein (long aliphatic chain) [38]. When the ethanol concentration was increased to 80%, 90%, and 99%, there was no significant change in the lutein content. Combined with the above results, the suitable ethanol concentration for the highest content of the lutein was considered 80%. Solvent-to-SP ratio was another parameter affecting the extraction of lutein; however, this parameter involves a process of finding the minimum volume to avoid the saturation limit of the lutein [39]. The lutein content increased as the ethanol-to-SP ratio increased (20–60 mL of ethanol per 1 g of spinach powder) (Figure 3B). The lutein content gave the highest value (2.21 mg/g of powder) when the ratio of SP to ethanol was 1:60. In addition, the content of lutein increased a little when the ratio of SP to ethanol surpassed 1:60. Combined with the above results, the suitable ratio for the highest content of the lutein was considered to be 1:60. The results of the lutein content extracted at different temperatures (25, 35, 45, 55, 65m and 75 °C) are tabulated in Figure 3C. The lutein content extracted from SP at 35 °C showed the highest value (2.16 mg/g of powder) out of the lutein content obtained via extraction temperatures at 25 °C and 45 °C. However, the lutein content decreased at temperatures above 55 °C, and at temperatures above 60 °C, the yield in lutein started to decline due to isomerization and degradation [38,40]. Hence, the extraction temperature of 35 °C was used in the subsequent experiments to determine the suitable extraction time. The results of the lutein content extracted at different times (0, 0.5, 1, 1.5, and 2 h) are illustrated in Figure 3D. With the increase in the extraction time from 0 to 0.5 h in the extraction system, the lutein content increased from 1.67 to 2.17 mg/g of powder. When the extraction time continued from 0.5 to 2 h, the lutein content was almost constant, probably because the lutein in SP was already sufficiently extracted [41]. Therefore, regarding our results, the optimum extraction parameters for lutein in SP were 80% ethanol for 0.5 h at 35 °C and with a 1/60 solvent-to-powder ratio.

### 3.6. Different Thermal/Non-Thermal Treatment Effects on Microbial Inactivation of SJ with SP-Superfine

Figure 4 shows the total aerobic bacteria (TAB) in SJ samples stored at 15 °C, 25 °C, and 35 °C for 25 days. The initial number of total aerobic bacteria in the SJ was 2.4 log CFU/mL, and when stored at 15 °C, the number of microorganisms reached 3.8 log CFU/mL in the untreated sample. The PEF- and thermal-treated SJ showed a number of microorganisms less than 1 log CFU/mL after treatment. The microbial growth stored at 15 °C was inhibited for 25 days in PEF-1, PEF-2, Thermal-1, and Thermal-2 (Figure 4A). When stored at 25 °C, the number of microorganisms reached 4.1 log CFU/mL in the untreated sample. The growth of microorganisms stored at 25 °C was inhibited for 25 days under all treatment conditions (Figure 4B). When stored at 35 °C, the number of microorganisms reached 5.0 log CFU/mL in the untreated samples, and microbial growth was inhibited for 25 days in all pasteurization treatments (Figure 4C). PEF physically damages the cell membrane, which releases the intracellular substances; thus, when the strengths of PEF exceed the critical value, vegetative cells are inactivated, or irreversible damage is induced. Similar results were found in Jin et al., (1999) [42], i.e., that the treatment of PEF (40 kV/cm) on cranberry juice significantly reduced the total aerobic bacteria by approximately 2 log cycles, and as the strengths of PEF increased, the TAB decreased (*p* < 0.05). Treatment of PEF (30 kV/cm, 150 pulse) decreased the total aerobic mesophilic count to about 2.7 log CFU/mL without a decrease in polyphenol contents [43]. And in Timmermans, Mastwijk [44], the TAB of samples treated with PEF were under the detection limit for 2 months at 4 °C, without deterioration of the initial quality, in orange juice. Furthermore, treatment of PEF at 18.8 kV/cm completely inactivated the aerobic bacteria in a fermented beverage by a reduction of approximately 3.7 log cycle; also, no growth was observed during 56 days at 4 °C [45]. Furthermore, PEF treatment on raspberry juice decreases the mold and yeast activities depending on the frequency. Also, PEF could more effectively inactivate the yeast than the mold [46]. However, it is difficult to inactivate microbial structures with a complex matrix because proteins, fats, and other compounds diminish the lethal effect of PEF; thus, the effects of PEF could be changed by the complexity of the beverage.

### 3.7. Lutein Contents and Antioxidant Activity of SJ Ring Storage

The lutein contents of SJ after pasteurization are shown in Figure 5. The initial contents of lutein in the SJ after PEF treatment (PEF-1, PEF-2) were 1.42 ± 0.01 mg/200 mL and 1.35 ± 0.02 mg/200 mL, and the untreated sample was 1.25 ± 0.00 mg/200 mL. In the present study, there was no decrease in lutein content via PEF treatment, and this result was similar to that of the previous study, in which PEF treatment inhibited the degradation of pigments in spinach extract [14]. However, the lutein contents of Thermal-1 were 0.93 ± 0.01 mg/200 mL, and Thermal-2 was not detected. These results were similar to the previous study, which reported that during the pasteurization (85 °C for 25 s) process, the lutein content was slightly decreased, and after sterilization (121.5 °C for 15 min), lutein was completely degraded and not detected [47]. Also, Chung, Leanderson [48] reported that regardless of the heating method, heat treatment causes a substantial loss of lutein in spinach. The lutein content of the untreated sample was 1.25 ± 0.00 mg/200 mL, which was higher than that of the heat-treated sample (0.93 ± 0.01 mg/200 mL) and lower than that of the PEF sample (1.42 ± 0.01, 1.35 ± 0.02 mg/200 mL, respectively) in the initial period at 15 °C; then, it significantly decreased during storage (*p* < 0.05). Although a reduction of lutein in SJ was found in all samples, lutein content was retained in the PEF-1 and PEF-2 rather than in Thermal-1 and Thermal-2. When stored at 35 °C, the untreated sample showed a complete decrease in lutein from day 20, and for PEF-2, a complete decrease in lutein occurred on day 25; thus, we observed a 5-days increase for preserving lutein content in SJ. Similar results were observed in Rios-Corripio et al., (2022) [45], wherein the sample treated with PEF (18.8 kV/cm) on a pomegranate-fermented beverage showed constant total phenol and flavonoid content during 56-day storage compared to other pasteurization technologies. Exposure to intense voltage could increase the antioxidant capacity due to the release of phenolic compounds; the treatment of PEF (7 kV/cm) on wines increases the bioactive compounds [49]. Furthermore, treatment of PEF effectively reduces the total aerobic bacteria without altering antioxidant compounds such as anthocyanin and the total phenolic contents in pomegranate juice [50]. In summary, it can be established that lutein contents were relatively stable in SJ treated with PEF, without the excessive degradation of lutein during storage observed in thermal treatment.

## 4. Conclusions

In this study, a jet mill was shown to efficiently produce microparticles (SP-superfine powder (Dv50 = 5.59 µm)). Due to the increase in specific surface area, the particle size reduction influences the hydration properties and oil holding capacity, which could affect the physicochemical properties and extraction of lutein. The optimum extraction parameters for lutein were 80% ethanol for 0.5 h at 35 °C, with a 1/60 solvent-to-spinach-powder ratio. As the size of the SP particles decreased, the antioxidant activities and lutein contents significantly increased due to the higher yield of lutein extraction (*p* < 0.05). SP-superfine presents the optimal extraction conditions for lutein, and 10 mL of the spinach extract has a lutein content of 11.24 mg. For the comparison of lutein contents, different thermal treatments on SJ were performed using PEF, and the results of PEF treatment had little effect on the lutein content compared to the untreated sample. However, the lutein content with thermal treatment decreased significantly from 4.46 ± 0.05 mg/200 mL to 3.34 ± 0.01 mg/200 mL (at 95 °C for 10 min) and was not detected (at 121 °C for 15 min). The lutein content of SJ decreased significantly with the increasing storage temperature during storage, but the loss of lutein in the PEF-treated group was relatively small compared to the thermal-treated group due to the higher contents in the initial storage period. Also, the PEF-treated sample at a field strength of 20 kV/cm and a specific energy of 70~150 kJ/L did not demonstrate a decrease in lutein content. Regarding all results in this study, we suggest that the combination of jet-milling and PEF technology could comprise a promising technology with which to produce lutein-enhanced SP juice with minimal lutein denaturation and less quality change during storage. However, further research should include the sensory properties of SJ. Also, the high cost and complicated conditions for operating PEF, as well as the lower scale of production, comprise the limitations of the application of PEF in the food industry.

## Figures and Tables

**Figure 1 foods-13-00834-f001:**
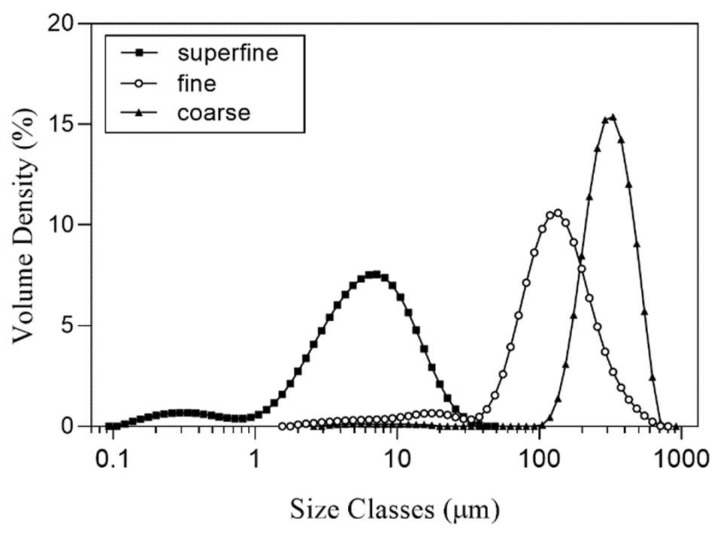
Particle size distribution of differently sized SP. SP-coarse: coarse spinach powder; SP-fine: fine spinach powder; SP-superfine: superfine spinach powder.

**Figure 2 foods-13-00834-f002:**
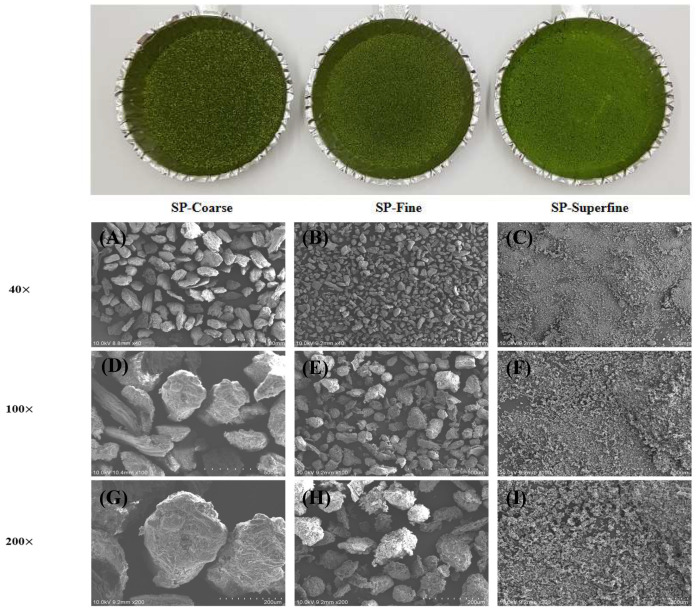
SEM images of differently sized SP. SP-coarse: coarse spinach powder; SP-fine: fine spinach powder; SP-superfine: superfine spinach powder. (**A**,**D**,**G**): SP-coarse; (**B**,**E**,**H**): SP-fine; (**C**,**F**,**I**): SP-superfine. (**A**–**C**) 40×; (**D**–**F**) 100×; (**G**–**I**) 200× magnification.

**Figure 3 foods-13-00834-f003:**
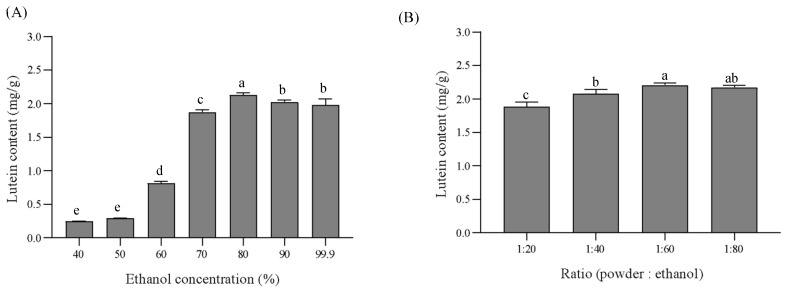
Lutein content of spinach by extraction conditions, ethanol concentration (**A**), solvent-to powder ratio (**B**), extraction temperature (**C**), and extraction time (**D**). Values with different letters within the same column (a–e) are significantly different (*p* < 0.05) according to the results of Duncan’s test.

**Figure 4 foods-13-00834-f004:**
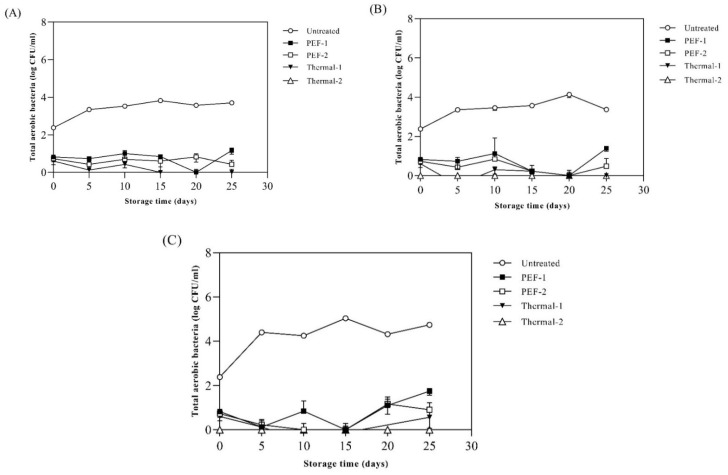
Effects of different thermal/non-thermal treatments on the total aerobic bacteria of spinach extract during storage at (**A**) 15 °C, (**B**) 25 °C, and (**C**) 35 °C. Untreated: no pasteurization; PEF-1: treated with 20 kV/cm 110 kJ/; PEF-2: treated with 20 kV/cm 150 kJ/L; Thermal-1: treated with 95 °C, 10 min; Thermal-2: treated with 121 °C, 15 min. Values are expressed as mean ± standard deviation (n = 5).

**Figure 5 foods-13-00834-f005:**
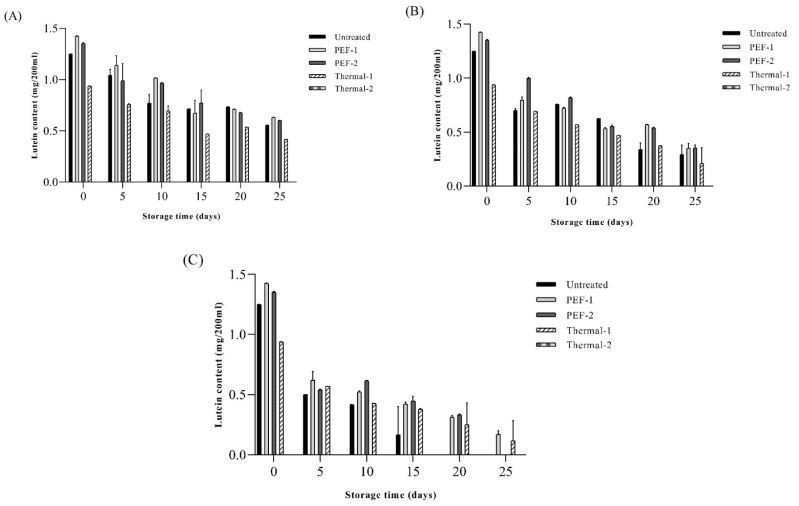
Effects of different thermal/non-thermal treatments on lutein content of spinach extract during storage at (**A**) 15 °C, (**B**) 25 °C, and (**C**) 35 °C. Untreated: no pasteurization; PEF-1: treated with 20 kV/cm 110 kJ/L; PEF-2: treated with 20 kV/cm 150 kJ/L; Thermal-1: treated with 95 °C, 10 min; Thermal-2: treated with 121 °C, 15 min. Values are expressed as mean ± standard deviation (n = 5).

**Table 1 foods-13-00834-t001:** Particle-size distribution and powder properties with differently-sized SP.

Samples	Particle-Size Distribution						Powder Properties Flowability		
	Dv_10_ (μm)	Dv_50_ (μm)	Dv_90_ (μm)	D[4,3] (μm)	Span	Specific Surface Area (m^2^/kg)	Bulk Density (g/mL)	Tap Density (g/mL)	Carr Index (%)
SP-coarse ^(1)^	191.60 ± 5.41 ^a,(2),(3)^	315.20 ± 3.37 ^a^	495.00 ± 5.15 ^a^	329.00 ± 6.44 ^a^	0.96 ± 0.03 ^c^	97.19 ± 4.11 ^b^	0.44 ± 0.00 ^a^	0.51 ± 0.00 ^b^	13.65 ± 0.28 ^c^
SP-fine	53.22 ± 2.84 ^b^	125.20 ± 1.30 ^b^	259.60 ± 4.45 ^b^	143.6 ± 2.07 ^b^	1.65 ± 0.04 ^b^	334.26 ± 11.33 ^b^	0.44 ± 0.00 ^a^	0.58 ± 0.00 ^a^	24.27 ± 0.59 ^b^
SP-superfine	1.59 ± 0.08 ^c^	5.59 ± 0.24 ^c^	14.16 ± 0.35 ^c^	6.93 ± 0.24 ^c^	2.24 ± 0.05 ^a^	8671.20 ± 414.26 ^a^	0.22 ± 0.00 ^b^	0.39 ± 0.00 ^c^	43.47 ± 0.49 ^a^

^(1)^ SP-coarse: coarse spinach powder; SP-fine: fine spinach powder; SP-superfine: superfine spinach powder. ^(2)^ Values are expressed as mean ± standard deviation (n = 5). ^(3)^ Values with different letters within the same column (a–c) are significantly different (*p* < 0.05) according to the results of Duncan’s test.

**Table 2 foods-13-00834-t002:** Hydration properties, lutein contents, and antioxidant activity with differently sized SP.

Samples	Hydration Properties			Antioxidant Activities		
	WHC (g/g)	SC (mL/g)	OHC (g/g)	Lutein Content (mg/g)	ABTS (%)	DPPH (%)
SP-coarse ^(1)^	4.55 ± 0.47 ^a,(2),(3)^	1.97 ± 0.06 ^a^	2.86 ± 0.01 ^c^	1.59 ± 0.02 ^b^	56.56 ± 2.01 ^b^	39.36 ± 1.94 ^a^
SP-fine	3.87 ± 0.08 ^b^	1.60 ± 0.10 ^b^	2.51 ± 0.03 ^b^	1.70 ± 0.10 ^b^	55.31 ± 0.31 ^b^	32.67 ± 1.36 ^b^
SP-superfine	3.26 ± 0.23 ^c^	1.33 ± 0.29 ^b^	2.30 ± 0.04 ^a^	2.09 ± 0.13 ^a^	65.01 ± 1.25 ^a^	36.82 ± 1.65 ^a^

^(1)^ SP-coarse: coarse spinach powder; SP-fine: fine spinach powder; SP-superfine: superfine spinach powder. ^(2)^ Values are expressed as mean ± standard deviation (n = 5). ^(3)^ Values with different letters within the same column (a–c) are significantly different (*p* < 0.05) according to the results of Duncan’s test.

## Data Availability

The original contributions presented in the study are included in the article, further inquiries can be directed to the corresponding author.

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
