# Peer review of "The Effects of Jet-Milling and Pulsed Electric Fields on the Preservation of Spinach Juice Lutein Contents during Storage"

_foods, 2024, doi:10.3390/foods13060834_

Round 1
Reviewer 1 Report
Comments and Suggestions for Authors
1. The chemical structural formula of carotenoids, including lutein, is best provided.
2. line 42-45. "Furthermore, to preserve the lutein contents during food processing, extraction is mainly applied rather than other food processing which includes heat but, the pasteurization process should proceed to produce juice product after extraction. " It is difficult to understand the correlation between this and the purpose of the experiment in this article. It should be carefully revised.
3. line 49, Please provide more accurate information, not just citing it.
4. line 52, Has jet milling technology ever been used to extract lutein from other raw materials or other compounds from spinach? Please add relevant introduction.
5. line 60, 1V? Is this the correct format for scientific research papers? Please confirm.
6. line 84, Are there any references or pre experimental results for the source of the parameters of the jet milling.
7. line 136, The title does not match the experimental content.
8. line 141, The detection method for lutein content should be placed where it is first used.
9.Table 1, The significance of liquidity indicators needs to be explained.
10. line 265. There is no corresponding test method.
11. line 298. format should be revised.
12. FIg. 4 should be revised carefully.
13. The limitation of this study should be added.
Comments on the Quality of English Language
English writing could be further updated
Author Response
|
No. |
question |
|
|
1 |
line 42-45. "Furthermore, to preserve the lutein contents during food processing, extraction is mainly applied rather than other food processing which includes heat but, the pasteurization process should proceed to produce juice product after extraction. " It is difficult to understand the correlation between this and the purpose of the experiment in this article. It should be carefully revised. |
Thank you for your comments. We revised this part to “Furthermore, to preserve the lutein contents during food processing, extraction is mainly applied but, the thermal process should proceed to increase the shelf life and decrease the risk of microbial contamination however, almost lutein in extract dramatically decreased due to the thermal process of pasteurization.” |
|
2 |
line 49, Please provide more accurate information, not just citing it.
|
Thank you for your comments. We revised this part to “A previous study has shown that the particle size reduction of raw materials preserves and improves the yield of lutein extraction that the lutein in superfine marigold powder (Dv50 = 10 μm) higher than coarse marigold powder (Dv50 = 315 μm) approximately 40%.” in line 51 . |
|
3 |
line 52, Has jet milling technology ever been used to extract lutein from other raw materials or other compounds from spinach? Please add relevant introduction. |
Thank you for your comments. However, there is no research using a jet mill for the extraction of lutein from spinach. Thus we added our research using the jet mill to increase the lutein contents from the marigold flower. |
|
4 |
line 60, 1V? Is this the correct format for scientific research papers? Please confirm. |
Thank you for your comments. We revised this part to “Treatment of PEF initiates the repulsion between the change-carrying molecules and forms the pore on the cell membrane which affects the microbial inactivation depending on the raw materials cellular structure, length, duration of field, and number of pulses [15].” in line 61. |
|
5 |
line 84, Are there any references or pre experimental results for the source of the parameters of the jet milling. |
Yes. We did our pre-experiment design according to research from our laboratory. (1) Muttakin, S., Kim, M. S., & Lee, D. U. (2015). Tailoring physicochemical and sensorial properties of defatted soybean flour using jet-milling technology. Food Chemistry, 187, 106-111. (2) Heo, T. Y., Kim, Y. N., Park, I. B., & Lee, D. U. (2020). Amplification of vitamin D2 in the white button Mushroom (Agaricus bisporus) by UV-B irradiation and jet-milling for its potential use as a functional ingredient. Foods, 9(11), 1713. (3) Kim, S. Y., Hong, S. Y., Choi, H. S., Kim, J. H., Jeong, S. H., Lee, S. Y., ... & Lee, D. U. (2023). Superfine Marigold Powder Improves the Quality of Sponge Cake: Lutein Fortification, Texture, and Sensory Properties. Foods, 12(3), 508.
|
|
6 |
line 136, The title does not match the experimental content. |
We change the title to “Investigation of Jet-milling and Pulsed electric fields effects on the preserving lutein contents of spinach juice during storage”. Thank you for your comments. |
|
7 |
line 141, The detection method for lutein content should be placed where it is first used. |
Yes. We explain the method in line 152. |
|
8 |
Table 1, The significance of liquidity indicators needs to be explained. |
The indicator related to liquidity is explained in line 2210, 212, and 222. |
|
9 |
line 265. There is no corresponding test method. |
The method of OHC, SC, and WHC is shown in line 111 and the method of lutein contents, ABTS, and DPPH is shown in line 152. |
|
10 |
line 298. format should be revised. |
Thank you for your comments, we revised this part carefully. |
|
11 |
Fig. 4 should be revised carefully. |
Thank you for your comments, we revised this part carefully. |
|
12 |
The limitation of this study should be added. |
Thank you for your comments. We revised and add the limitation of this study in line 472-473. “However, further research should be including sensory properties of SJ. Also, the high cost and complicate conditions for operating PEF and lower scale of production is the limitation of application PEF in food industry.” |

Reviewer 2 Report
Comments and Suggestions for Authors
The article investigates the effects of jet-milling and pulsed electric fields on the lutein content and antioxidant activity of spinach powder and juice. The article is well-written and provides a clear overview of the objectives, methods, results, and discussion of the study. However, the article could be improved by addressing the following comments and suggestions:
The introduction is well-written and provides a clear overview of the topic and the objectives of the study. However, it could be improved by adding some citations to support the statements and claims made in the text. For example, you could cite some sources that show the benefits of lutein for eye health, the factors that affect the stability and bioavailability of lutein, and the previous studies that used jet-milling and pulsed electric fields for food processing. The introduction could also be more concise and avoid repeating the same information. For example, you could merge the sentences that mention the extraction and pasteurization processes, and the sentences that describe the chemical structure and properties of lutein.
The discussion section should provide a clear and logical explanation of the findings, based on the relevant literature and theories. For example, you could explain why the particle size reduction of SP affects the hydration properties and OHC, and why the ethanol concentration and solvent ratio affect the lutein extraction yield. You could also discuss the advantages and disadvantages of using JM and PEF for preserving the lutein content of SJ, and compare your results with previous studies that used similar or different methods. You could also mention the limitations of your study, such as the sample size, the experimental design, and the measurement errors, and suggest some directions for future research.
See the attachment for more comments.

Author Response
|
No. |
question |
|
||||||
|
1 |
Title: (1) superfine spinach powder, what you mean by superfine (2) Non-thermal pasteurization, you can remove this part |
Thank you for your comment. We revised the title to “Investigation of Jet-milling and Pulsed electric fields effects on the preserving lutein contents of spinach juice during storage”. |
||||||
|
2 |
Line 28: do not repeat |
Thank you for your comment. We deleted the repeated part.
|
||||||
|
3 |
line 72, How you select this parameter
|
Fig.1. The absorbance of spinach juice with various PEF conditions.
According to the results, we selected 2 conditions for experiments to perform this research.
The pasteurization on conditions selected by described in food regulation in Korea. |
||||||
|
4 |
line 125, This is sterilization rather than pasteurization |
Regarding your comments, we revised the pasteurization to thermal treatment which includes the pasteurization and sterilization conditions in overall manuscript. |
||||||
|
5 |
line 126, I do not think it falls under pastuerization |
Thank you for your comment. Regarding your comments, we revised this condition to sterilization conditions. Thank you for your comment. |
||||||
|
6 |
line 227, which powder |
We added the powder name in line 282. “than coarse and fine mushroom powder” |
||||||
|
7 |
line 257, particle of which product |
We added the powder name in line 285. “between the defatted soybean flour particles” |
||||||
|
8
|
line 334, you should refer to recent literature Line 339, You can find so many studies on juice |
Thank you for your comment. We revised this part in line 377 and line 382. (1) Mukhtar, K., et al., Potential impact of ultrasound, pulsed electric field, high-pressure processing, microfludization against thermal treatments preservation regarding sugarcane juice (Saccharum officinarum). Ultrasonics Sonochemistry, 2022: p. 106194. (2) Buitimea-Cantúa, G. V., Rico-Alderete, I. A., Rostro-Alanís, M. D. J., Welti-Chanes, J., Escobedo-Avellaneda, Z. J., & Soto-Caballero, M. C. (2022). Effect of High Hydrostatic Pressure and Pulsed Electric Fields Processes on Microbial Safety and Quality of Black/Red Raspberry Juice. Foods, 11(15), 2342. In line 377, we added “Treatment of PEF (30 kV/cm, 150 pulse) decrease the total aerobic mesophilic count about 2.7 log CFU/mL without decrease in polyphenol contents [42]”. In line 383, we added “Furthermore, treatment of PEF on raspberry juice decreases the mold and yeast activities depending on the frequency also, PEF could effectively inactivate the yeast than mold [45]”. |
||||||
|
9 |
line 366. What’s the limit of the safe heating range |
The range in which lutein is not detected is the limit. |

Reviewer 3 Report
Comments and Suggestions for Authors
MATERIALS AND METHODS
The authors use many variables and do not make the non-variable parameters clear enough. It would be very convenient to include in this section a diagram of the experimental design, indicating the number of samples and repetitions carried out in each experiment. In general, in this section, some methods are explained in too much detail and others in very little detail. In many cases, the units in which the results are expressed are not indicated.
Examples:
76-86.- In this section, the authors do not indicate the number of samples they analyze. Are all the leaves ground together? That would not be correct since, in that case, only one sample would be analyzed.
93- 107. Physical properties of spinach powder (SP)
How many samples were taken? How many repetitions were done? Explain all calculations using formulas and quotes.
132-135.- How many samples were taken from each temperature?
157.- 2.8. Determination of antioxidant activities
In this section the authors must indicate the units in which the results are expressed.
RESULTS AND DISCUSSION
In general, the work is very messy and there is a need to spend time writing it with more interest, including a more in-depth discussion.
Examples:
1.- Section 3.5 should come after 3.2, since the data discussed in section 3.5 is in table 2.
265.- 3.4. Optimization of lutein extraction conditions
The figures are of poor quality.
Non-variable conditions must be clearly indicated in each figure.
CONCLUSIONS
It should be rewritten since it includes results and discussion.
Author Response
|
No. |
question |
|
|
1 |
Line 76-86.- In this section, the authors do not indicate the number of samples they analyze. Are all the leaves ground together? That would not be correct since, in that case, only one sample would be analyzed. |
Since there is a great difference in the size and shape of spinach, leaves with similar size and shape were selected and applied to the experiment to obtain a certain experimental value. In the experiment, 8 to 10 grams of spinach leaves were used. Used separately. The selected spinach is placed in a PEF chamber and then treated with a pulsed electric field according to the given conditions. Then, using the standard extraction method of the food industry, the absorbance was measured at 446 nm. |
|
2 |
Line 93- 107. Physical properties of spinach powder (SP) How many samples were taken? How many repetitions were done? Explain all calculations using formulas and quotes. |
The 8-10 g of spinach powder was measured 5 times. We explain the detailed equation in lines 115-117. WHC (g/g) = (Weight of sample (g))/(Weight of sediment (g)) (1) SC (mL/g) = (Volume occupied by sample (mL))/(Weight of dry sample (g)) (2) OHC (g/g) = (Weight of sediment (g)-Dry weight (g))/(Weight of sample (g)) (3)
|
|
3 |
Line 132-135.- How many samples were taken from each temperature? |
Five samples of 200 ml each were collected. Regarding your comment, we added this part in line 133. “For each group, 5 samples were measured after put into a glass bottle and homogenized for 3 min at 10,000 rpm using a homogenizer (T18 ultra-turra, IKA, Germany) before the experiment.” |
|
4 |
Line 157.- 2.8. Determination of antioxidant activities In this section the authors must indicate the units in which the results are expressed |
Thank you for your comment. We revised this part. We added “And the units of ABTS and DPPH are percent.” in line 189. |
|
5 |
1.- Section 3.5 should come after 3.2, since the data discussed in section 3.5 is in table 2. |
Thank you for your comment. We revised this part. |
|
6 |
265.- 3.4. Optimization of lutein extraction conditions |
Thank you for your comment. We revised this part. |
|
7 |
Non-variable conditions must be indicated in each figure. |
Thank you for your comment. We revised this part. |
|
8 |
Conclusions: it should be rewritten since it includes results and discussion. |
Thank you for your comment. We revised this part. “In this study, a Jet mill could efficiently produce microparticles (SP-superfine powder (Dv50 = 5.59 µm)). Due to the increase in specific surface area, particle size reduction influences the hydration properties and oil holding capacity which could affect the physico-chemical properties and extraction of lutein. The optimum extraction parameters for lutein were 80 % ethanol, for 0.5 h at 35 °C and with 1/60 solvent to spinach powder ratio. As the size of the SP particle decreased, the antioxidant activities and lutein contents significantly increased due to the higher yield of lutein extraction (p < 0.05). SP-superfine and optimal extraction conditions of lutein, and the 10 mL of the spinach extract have an 11.24 mg lutein content. For the comparison of lutein contents, different thermal treatment on SJ was performed using PEF and the results of PEF treatment had little effect on lutein content compared to the untreated sample. However, the lutein content of thermal treatment decreased significantly from 4.46 ± 0.05 mg/200 mL to 3.34 ± 0.01 mg/200 mL (at 95 ℃ for 10 min) and was not detected (at 121 ℃ for 15 min). SP-superfine and optimal extraction conditions of lutein, and the 10 mL of the spinach extract have an 11.24 mg lutein content. The lutein content of SJ decreased significantly with increasing storage temperature during storage, but the loss of lutein in the PEF-treated group was relatively smaller than in the thermal-treated group due to the higher contents at initial storage period. Also, PEF treated sample at field strength of 20 kV/cm and specific energy of 70 ~ 150 kJ/L was not decrease in lutein content. Regarding all results in this study, we suggest that the complex of Jet-milling and PEF technology could be the promising technology to produce lutein-enhanced SP juice with minimal lutein denaturation and less quality change during storage. However, high cost and complicate conditions for operating PEF and lower scale of production is the limitation of application PEF in food industry.”
|

Round 2
Reviewer 1 Report
Comments and Suggestions for Authors
Ths manuscirpt was improved
Comments on the Quality of English Language
Ths manuscirpt was improved
Author Response
Thank you for your comments.
Reviewer 3 Report
Comments and Suggestions for Authors
The article can be improved.
For example:
It would be very convenient to include in this section a diagram of the experimental design, indicating the number of samples and repetitions carried out in each experiment.
265.- 3.4. Optimization of lutein extraction conditions
The figures are of poor quality.
Author Response
|
No. |
question |
|
|
1 |
It would be very convenient to include in this section a diagram of the experimental design, indicating the number of samples and repetitions carried out in each experiment. |
|
|
2 |
265.- 3.4. Optimization of lutein extraction conditions. The figures are of poor quality. |
We revised this part. Thank you for your comments. |
